

# Geometric constructions of generalized dual-unitary circuits from biunitarity

**Michael A. Rampp, Suhail A. Rather and Pieter W. Claeys**

Max Planck Institute for the Physics of Complex Systems, 01187 Dresden, Germany

## Abstract

We present a general framework for constructing solvable lattice models of chaotic many-body quantum dynamics with multiple unitary directions using biunitary connections. We show that a network of biunitary connections on the kagome lattice naturally defines a multi-unitary circuit, where three 'arrows of time' directly reflect the lattice symmetry. These models unify various constructions of hierarchical dual-unitary and triunitary gates and present new families of models with solvable correlations and entanglement dynamics. Using multilayer constructions of biunitary connections, we additionally introduce multilayer circuits with monoclinic symmetry and higher level hierarchical dual-unitary solvability and discuss their (non-)ergodicity. Our work demonstrates how different classes of solvable models can be understood as arising from different geometric structures in spacetime.

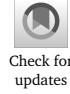
doi:10.21468/SciPostPhys.18.6.182

# 1 Introduction

Recent years have seen a surge of progress on exact solutions of the dynamics of certain interacting many-body quantum systems. Dual-unitary circuits are useful examples of solvable yet chaotic models, in which the dynamics of correlation functions, entanglement, and scrambling could be exactly characterized [1–6]. Following these initial results, dual-unitary models have gained intense attention both theoretically and experimentally [7–28]. These models are characterized by an underlying symmetry between space and time which, as well as leading to solvability, makes certain aspects of their dynamics pathological. For example, dual-unitary circuits exhibit the maximal entanglement growth consistent with locality [5,8,29], thermalize exactly for quenches from a large class of states [8], display a maximal butterfly velocity [3], and dynamical correlations vanish almost everywhere except on the edge of the causal light cone [2]. In an attempt to relax (some of) these pathologies while preserving the solvability, the zoo of exactly solvable models has since been extended by a variety of models that can broadly be classified as 'multi-unitary' [30–37]. These models include triunitary circuits [31] and lattices with honeycomb-like space-time symmetries [36], both of which give rise to three 'arrows of time'. The latter models can be reinterpreted as a special case of hierarchical dual-unitary models, which present a systematic way of relaxing the dual-unitary condition [32]. Despite the similar physics exhibited by such hierarchical dual-unitary models and triunitary ciruits, their underlying constructions appear seemingly unrelated.

Moving to extend dual-unitarity in a complementary direction, Ref. [38] introduced a graphical calculus that allowed to rewrite different classes of models with dual-unitary-like properties in a unified manner and extend the notion of dual-unitarity to biunitarity [39]. Originating from quantum information and category theory, this graphical calculus is based on 'biunitarity connections', objects which satisfy different notions of unitarity. Biunitarity provides a framework in which dual-unitary circuits and so-called dual-unitary interactions round-a-face can be treated in a uniform manner. Such interactions round-a-face exhibit the same physics as dual-unitary circuits while being seemingly unrelated [30], but can be understood as different realizations of the same biunitary spacetime lattice. This connection suggests a similar relation between triunitary models and hierarchical dual-unitary models, which we here explore.

In this work, we apply the framework of biunitarity to generalized dual-unitary circuits. First, we use biunitarity to unify and extend various constructions generalizing dual-unitarity. We show how known triunitary and hierarchical dual-unitary constructions both arise from biunitarity connections arranged on the kagome lattice. Operator dynamics inherits the space-time symmetry of this lattice, leading to the three characteristic light cones and resulting in solvable entanglement dynamics. Second, we show how gates with different entangling prop-

erties can be systematically constructed. Hierarchically generalized dual-unitary gates, while leading to similar correlation dynamics, can have different entanglement dynamics set by the entangling properties of the underlying gates, which give rise to different entanglement velocities (to be contrasted with the maximal entanglement velocity of dual-unitary gates [5,8,40]). These constructions use a property of biunitary constructions that is hitherto unexplored in the literature on many-body quantum dynamics: The possibility to layer different constructions and in this way tune the entangling properties of the gate. Taken together, these results make explicit how different geometric structures in spacetime can give rise to different classes of solvable many-body quantum dynamics.

## 2 Dual-unitarity and its generalizations

### 2.1 Dual-unitarity

Unitary circuits [41] are models of many-body quantum dynamics which – in the spirit of classical cellular automata – aim to capture the essential features of quantum dynamical systems: locality and unitarity. Because of their gate-based nature, unitary circuits can directly be implemented on quantum computing platforms. Conversely, they naturally appear when discretizing Hamiltonian evolution via the Trotter-Suzuki decomposition [42,43].

We consider a chain of $q$-level systems (qudits) that may be operated on pairwise with a unitary gate $U$ graphically depicted by a blue box

$$
U_{ab,cd} = \vcenter{\hbox{\includegraphics{}}} , \qquad U^\dagger_{ab,cd} = \vcenter{\hbox{\includegraphics{}}} . \tag{1}
$$

These are both $q^2 \times q^2$ complex matrices, where the Hermitian conjugate is depicted by a red box. We adapt tensor network notation: Tensors with $k$ indices are represented by diagrams with $k$ unconnected lines, also referred to as wires, and connecting two lines implies the contraction of the corresponding index. The time evolution operator of the full chain is then constructed by connecting copies of the two-site gate $U$ in a brickwork fashion

$$
\mathbb{U}^t = \vcenter{\hbox{\includegraphics{}}} \tag{2}
$$

The total number of discrete time steps corresponds to the number of layers in this circuit (here $t = 2$).

This evolution operator is said to be dual-unitary when the constituting gates are unitary in both the vertical and the horizontal direction. Specifically, these gates satisfy

$$
\vcenter{\hbox{\includegraphics{}}} = \Big| \; \Big| , \qquad \vcenter{\hbox{\includegraphics{}}} = \Big) , \qquad \vcenter{\hbox{\includegraphics{}}} = \Big( . \tag{3}
$$

The first equality describes the standard (vertical) unitarity of the gate, whereas the second and third equalities result in additional unitarity along the horizontal direction. Remarkably, dual-unitarity of the gates guarantees that the many-body dynamics described by the full circuit is exactly solvable. As two specific predictions, the spatiotemporal correlations in dual-unitary dynamics vanish everywhere except on the edge of the causal light cone [2], leading to correlations spreading with maximal velocity, and these circuits exhibit maximal entanglement growth with entanglement velocity $v_E = 1$ [5, 8, 40].

## 2.2 Triunitarity and hierarchical dual-unitarity

Multiple works extended the notion of dual-unitarity in different directions, leading to richer dynamics while preserving their solvability [30–37]. Shortly after the introduction of dual-unitarity, Ref. [31] introduced the notion of *triunitarity*. The building blocks of triunitary circuits are three-site unitary gates, which satisfy unitarity in three different directions. Triunitary circuits are graphically represented as

$$\mathbb{U}^t = \ldots \qquad \ldots \tag{4}$$

with the gates satisfying additional unitarity in directions rotated by $\pm\pi/3$ from the usual arrow of time

$$\tag{5}$$

The resulting dynamics of correlation functions reflects the space-time symmetry of these circuit in the appearance of three different light rays at $x = 0$ and $x = \pm t$. These can be contrasted with the two light rays in dual-unitary circuits at $x = \pm t$. Next to correlation functions, the entanglement dynamics of triunitary circuits can also be characterized exactly. While the early-time growth of entanglement is universal, with the entanglement velocity being maximal among circuits composed of three-site gates, the entanglement dynamics for times greater than half the subsystem size depends on the microscopic structure of the gate. In contrast to dual-unitary circuits, the calculation of indicators of random-matrix like behavior has not been accomplished.

Similar progress was made with the introduction of *hierarchically generalized dual-unitarity* [32]. It is possible to generalize the dual-unitary condition (3) to include multiple gates, where

the simplest extension reads

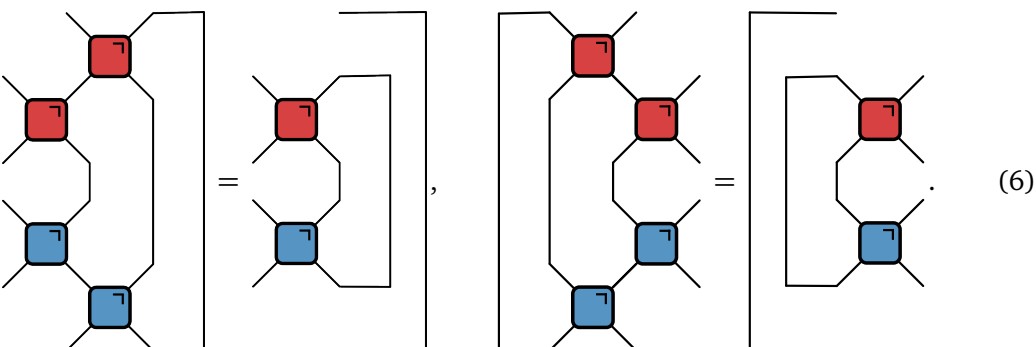

$$(6)$$

The resulting gates are known as DU2 gates. It was found that such generalized dual-unitary circuits exhibit a non-maximal entanglement velocity [34, 35]. Nevertheless, in the simplest generalization correlators are still supported exclusively on one-dimensional rays in spacetime [32], again given by $x = 0$ and $x = \pm t$, and the entanglement line tension, a function characterizing the entanglement dynamics on large wavelengths, is piecewise linear [34, 35]. Another motivation to study generalized dual-unitary circuits stems from the observation that the CNOT gate, a paradigmatic example of a DU2 gate, can be interpreted as a driven version of the kinetically constrained east model [33, 44–48]. For these DU2 gates the spacetime structure of the correlations is the same as for triunitary circuits, suggesting a connection between these conditions.

Higher levels of the hierarchy are defined by extension of the condition (6) to contain a contraction of $k$ gates along the diagonal. For instance, for $k = 3$ the condition reads

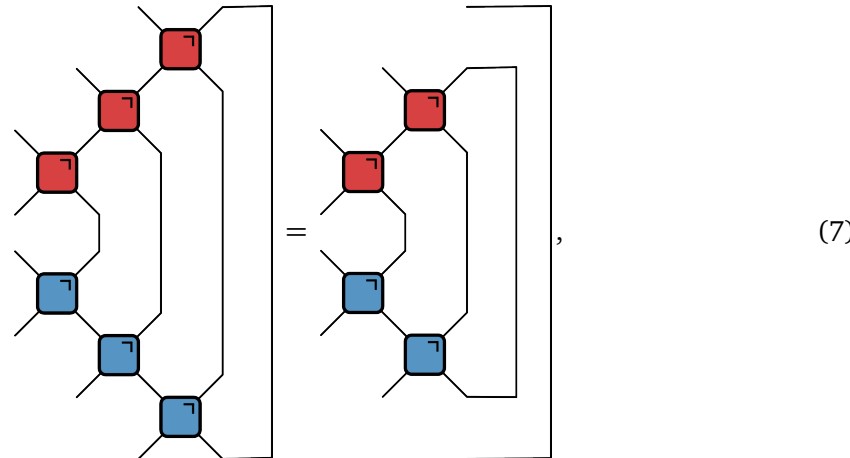

$$(7)$$

(plus an analogous mirrored condition) and the resulting gates are called DU3 gates. For higher levels of the hierarchy the restriction of a piecewise linear entanglement line tension is lifted in a region of spacetime, but at the cost of solvability in this region [35]. While DU2 gates have been the subject of various recent studies [32–37], DU3 gates are not that well-studied, largely due to the absence of constructions leading to nontrivial dynamics.

Dual-unitary gates are uniquely defined by fixing the entanglement velocity to be maximal, $v_E = 1$ [29]. Hierarchical dual-unitarity similarly imposes constraints on the entanglement velocity, which is quantized in DU2 circuits. The entanglement velocity $v_E$ can be expressed in terms of the Schmidt rank $\mathcal{R}$ of the two-site gate, which has a flat Schmidt spectrum, as [34, 35]

$$v_E = \frac{\log \mathcal{R}}{\log q^2} . \qquad (8)$$

Following Ref. [49], the Schmidt rank is defined as the number of nonzero singular values in the operator-Schmidt decomposition of the two-site gate $U$, i.e., in the decomposition

$$U = \sum_{i=1}^{\mathcal{R}} \lambda_i X_i \otimes Y_i, \quad \text{with} \quad \text{Tr}(X_i^\dagger X_j) = \text{Tr}(Y_i^\dagger Y_j) = \delta_{ij}. \tag{9}$$

Flatness of the Schmidt spectrum additionally fixes $\lambda_i = q/\sqrt{\mathcal{R}}, \forall i = 1 \ldots \mathcal{R}$. Since $\mathcal{R}$ is an integer $\mathcal{R} \in \{1, 2, \ldots, q^2\}$, $v_E$ can only take certain discrete values for each Hilbert-space dimension $q$. It was recently conjectured (and proven for the special case of permutation gates) [35] that the only non-trivial Schmidt rank for DU2 gates with $q$ prime is $\mathcal{R} = q$, corresponding to $v_E = 1/2$. In higher levels of the hierarchy, entanglement dynamics remains largely unexplored. While the restricted solvability prohibits the direct calculation of the entanglement velocity, it is nevertheless possible to bound the entanglement velocity from the light-cone dynamics [35].

## 2.3 Biunitarity and the shaded calculus

Dual-unitary gates can also be seen as a specific realization of a *biunitary connection* [39,50]. Originating in the theory of subfactors [51,52] and having found uses in quantum information and category theory, biunitary connections are algebraic objects satisfying two notions of unitarity, typically referred to as 'vertical' and 'horizontal' unitarity [50]. Examples include dual-unitary gates and dual-unitary interactions round-a-face, but also complex Hadamard matrices, quantum Latin squares, and unitary error bases. All such objects have found various applications in quantum information, e.g. in quantum error correction and teleportation [53–55], and their use in quantum many-body dynamics is being increasingly investigated [7,35,39,56–62].

   While these biunitary connections are all seemingly distinct objects, they can be represented in a unified way through the use of the *shaded calculus* [50,52]. This calculus extends the graphical language of tensor network diagrams to also include shaded regions. Tensors are still represented by shapes, but indices can now be represented as either wires or shaded regions. Every tensor is indexed by both the wires connected to it and the regions bordering it, and tensor contraction as a summation of indices corresponds to either connecting wires or closing regions. As one illustration of the shaded calculus, consider the following diagram:

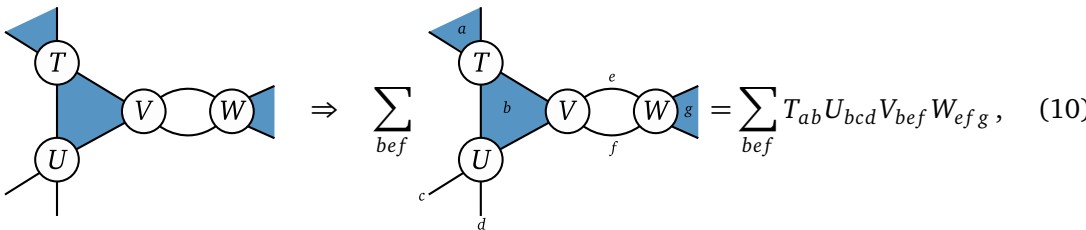

where we have explicitly written out all indices and summations on the right-hand side.

   The different biunitaries take a simple form in this graphical calculus, corresponding to either of the following diagrams:

$$\text{(11)}$$

Each biunitary $U$ has a Hermitian conjugate $U^\dagger$, which we again denote with a red background,

and these satisfy a set of graphical identities corresponding to vertical unitarity,

$$\tag{12}$$

and horizontal unitarity

$$\tag{13}$$

In the above diagrams, every region can either be shaded or unshaded (as indicated by the transparent background), returning the different conditions on the different biunitaries. The choice of shading pattern determines the specific choice of biunitary connection and the corresponding unitarity conditions. Within this shaded calculus, these biunitaries satisfy the same graphical identities as dual-unitary gates, such that any graphical derivation for dual-unitary dynamics hence directly extends to corresponding biunitary dynamics.

For a detailed introduction to biunitarity and the shaded calculus we refer the reader to Refs. [39, 50]. Ref. [50] introduces the notion of biunitarity, develops the shaded calculus, and shows how different biunitaries can be combined to return more complicated biunitaries. Any arrangement of biunitary connections can be represented as a quantum circuit, where the shaded regions typically introduce additional control parameters in the unitary circuits. Ref. [39] studies biunitarity in the context of many-body quantum dynamics and presents an exhaustive list of biunitary connections and their realizations as quantum circuits. These circuits all share the same properties of dual-unitary gates, e.g. again leading to light-cone correlation dynamics and a maximal entanglement velocity $v_E = 1$. Here we simply list the relevant objects, their properties, and the representation as a unitary gate.

**Dual-unitary gates.** In the absence of shading, biunitaries return dual-unitary gates. The shaded calculus returns the usual tensor network representation,

$$\tag{14}$$

$$= U_{ab,cd} \ ,$$

and vertical and horizontal unitarity imply the unitarity in the time and space direction respectively. Unitarity is expressed as

$$\sum_{c,d} U_{ab,cd} U^{\dagger}_{cd,ef} = \delta_{ae}\delta_{bf} \,, \tag{15a}$$

$$\sum_{c,d} c \left( \begin{matrix} a & b \\ U \\ U^{\dagger} \\ e & f \end{matrix} \right) d \;=\; \begin{matrix} a & b \\ \\ \\ e & f \end{matrix} \,, \tag{15b}$$

where we have made the graphical notation explicit, and spatial unitarity is expressed as

$$\sum_{b,d} U_{ab,cd} U^{*}_{be,df} = \delta_{ae}\delta_{cf} \,, \tag{16a}$$

$$\sum_{b,d} \begin{matrix} a & \quad b \quad & e \\ & U \qquad U^{*} & \\ c & \quad d \quad & f \end{matrix} \;=\; \begin{matrix} a & \rule{3cm}{0.4pt} & e \\ & & \\ c & \rule{3cm}{0.4pt} & f \end{matrix} \,. \tag{16b}$$

**Complex Hadamard matrices.**  A complex Hadamard matrix (CHM) is a $q \times q$ matrix $H$ that is proportional to a unitary matrix and with all matrix elements having unit modulus, i.e. $|H_{ab}| = 1, \forall a, b$, which fixes $H^{\dagger}H = HH^{\dagger} = q\mathbb{1}$. Biunitaries with two opposing shaded regions correspond to complex Hadamard matrices [52]:

$$\begin{matrix} a \\ H \\ b \end{matrix} \;=\; \begin{matrix} a \\ \square \\ b \end{matrix} \;=\; H_{ab} \;. \tag{17}$$

Depending on the orientation of these connections in a quantum circuit, complex Hadamard matrices define either single-site unitary gates or two-site controlled phase gates.

Vertical unitarity fixes these matrices to be proportional to unitary matrices

$$\sum_{b} H_{ab} H^{\dagger}_{bc} \propto \delta_{ac} \,, \tag{18a}$$

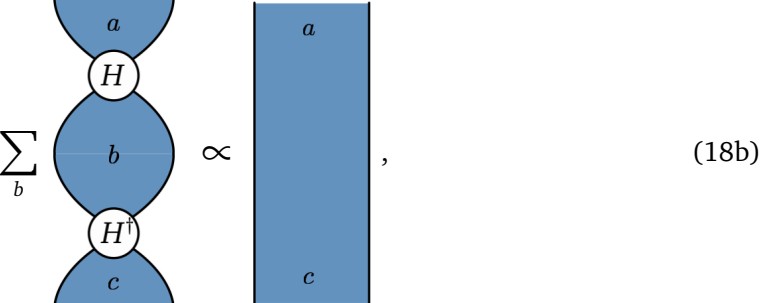

$$\sum_{b} \;\propto\; \,, \tag{18b}$$

and horizontal unitarity fixes all matrix elements to have unit modulus

$$H_{ab}H_{ab}^* = 1, \tag{19a}$$

$$\vcenter{\hbox{\includegraphics{eq19b}}} \tag{19b}$$

The above identity illustrates the rule that multiple regions can be joined without being closed, in contrast to wires. In this case, all indices of tensors bordering the joint region are fixed to the same value, but not summed over.

**Dual-unitary interactions round-a-face.** Dual-unitary interactions round-a-face (DUIRF), also known as quantum crosses, are set of unitary $q \times q$ matrices $\{F_{ab}|a, b = 1 \ldots q\}$ with matrix elements $(F_{ab})_{cd}$, such that the matrices $\tilde{F}_{cd}$ with matrix elements $(\tilde{F}_{cd})_{ab} = (F_{ab})_{cd}$ are also unitary [30]. Quantum crosses correspond to fully shaded biunitaries [39].

$$\vcenter{\hbox{\includegraphics{eq20}}} = (F_{ab})_{cd}. \tag{20}$$

Vertical unitarity fixes the unitarity of all $F_{ab}$

$$\sum_d (F_{ab})_{cd}(F_{ab})_{ed}^* = \delta_{ce}, \tag{21a}$$

$$\sum_d \vcenter{\hbox{\includegraphics{eq21b}}}, \tag{21b}$$

whereas horizontal unitarity fixes the unitarity of all $\tilde{F}_{cd}$

$$\sum_b (\tilde{F}_{cd})_{ab}(\tilde{F}_{cd})_{eb}^* = \delta_{ae}, \tag{22a}$$

$$\sum_b \vcenter{\hbox{\includegraphics{eq22b}}}. \tag{22b}$$

These matrices can be interpreted as one-site unitary gates controlled by two neighbors, where we can either fix $a$ and $b$ as control parameters to obtain a one-site unitary gate in the time direction, or fix $c$ and $d$ as control parameters to obtain a one-site unitary gate in the spatial direction.

**Unitary error bases.** A unitary error basis (UEB) is a complete orthogonal family of $q \times q$ unitary matrices $\{V_a | a = 1 \ldots q^2\}$, where orthogonality is with respect to the Frobenius norm, i.e. $\text{Tr}[V_a^\dagger V_b] = q \delta_{ab}$ [53]. Biunitaries with a single shaded region correspond to a unitary error basis [63, 64]:

$$\begin{array}{c} \includegraphics \end{array} \quad = \quad a - \fbox{} \quad = (V_a)_{bc} \; . \tag{23}$$

Vertical unitarity implies the unitarity of each matrix $V_a$, such that we can interpret these as controlled single-site unitary gates, and horizontal unitarity fixes the completeness and orthonormality. Since UEBs will not be the main focus of this work, we leave the correspondence between horizontal and vertical unitarity and these properties implicit and refer the reader to Refs. [39, 50] for a detailed discussion. A famous example of a unitary error basis is the family of Pauli matrices.

**Quantum Latin squares.** A quantum Latin square (QLS) is a $q \times q$ grid of states in a $q$-dimensional Hilbert space, $\{Q_{ab} | a, b = 1 \ldots q\}$, such that each row and each column of $Q$ form a complete orthonormal basis for this Hilbert space [65]. Biunitaries with two adjacent shaded regions define quantum Latin squares [52].

$$\begin{array}{c} \includegraphics \end{array} \quad = \quad a - \fbox{} \quad = (Q_{ab})_c \; . \tag{24}$$

Vertical and horizontal unitarity fix orthonormality and completeness of the rows and columns respectively. These gates can be interpreted as a family of controlled single-site gates, where we can take either $a$ or $b$ to be the control parameter, and the corresponding column/row of $Q$ fixes the basis transformation. We again refer the reader to Refs. [39, 50] for a detailed discussion.

## 3 Biunitary connections on the kagome lattice

In the following section, we show that arranging biunitary connections on the kagome lattice gives rise to a general class of circuits with solvability akin to generalized dual-unitary circuits of the DU2 type and triunitary circuits. This class unifies several existing constructions and shows how DU2-type solvability arises from a geometric description. We then use this framework to give several new constructions, e.g., based on unitary error bases, quantum Latin squares, and introduce a triunitary "interactions-round-a-face" circuit. We show that for constructions admitting a formulation as a brickwork unitary circuit, the underlying gates satisfy the DU2 condition. Moreover, we clarify the relationship between triunitarity and generalized dual-unitarity.

The kagome lattice is constructed from corner-sharing equilateral triangles. It is an Archimedean lattice where at each vertex two triangles and two hexagons meet. A unitary evolution operator is obtained by placing a biunitary connection at each vertex of the kagome lattice and

orienting it in the following way to ensure unitarity:

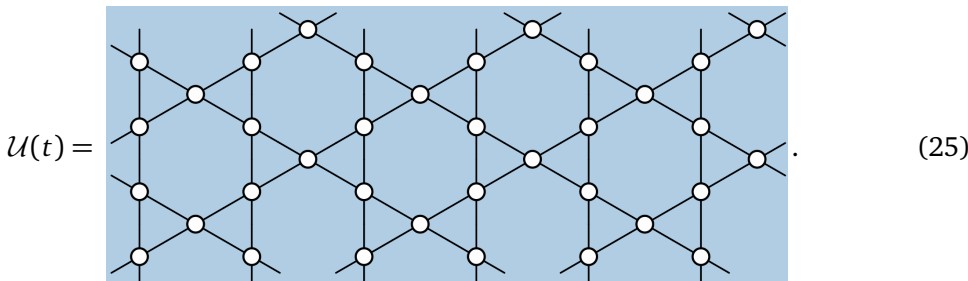

$$\mathcal{U}(t) = \qquad\qquad\qquad . \qquad (25)$$

Here every face can either be shaded or unshaded as long as overall consistency of Hilbert space dimensions is preserved. Time goes upward and the depicted circuit corresponds to $t = 2$ time steps (or layers).

## 3.1 Correlation functions

We will first show how, for any shading pattern, the dynamical correlation functions of local operators can be exactly obtained. Furthermore, they are supported along three rays in spacetime, $x = 0$ and $x = \pm t$, just as in DU2 and in triunitary circuits. We focus on traceless operators that are initially localized at one site only. The operator evolves in time as $\sigma(t) = \mathcal{U}(t)\sigma(0)\mathcal{U}(t)^{\dagger}$. In an infinite translationally invariant system there are four inequivalent choices of sites for the placement of the operator. The first choice is

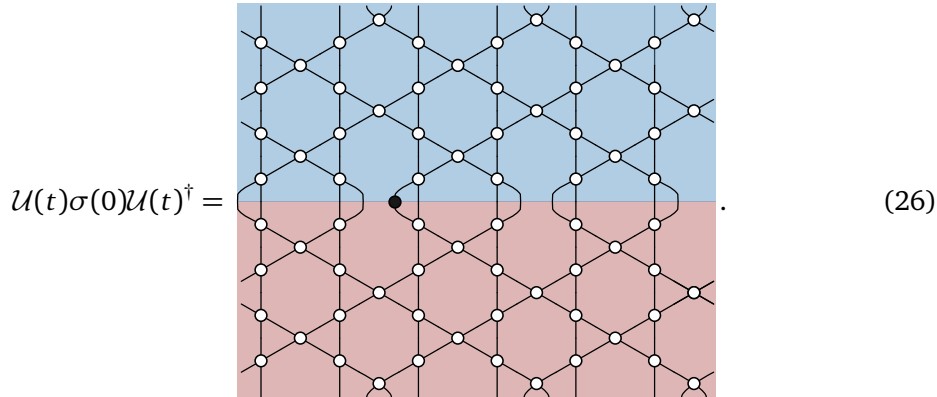

$$\mathcal{U}(t)\sigma(0)\mathcal{U}(t)^{\dagger} = \qquad\qquad\qquad . \qquad (26)$$

Here, red shading denotes the Hermitian conjugate. The operator $\sigma(0)$ can correspond to either a single-site operator acting on the Hilbert space corresponding to the wire, a single-site operator acting on a single shaded region, or a two-site operator acting on two neighboring shaded regions. The specific interpretation will depend on the choice of shading pattern.

Using vertical unitarity, Eq. (26) can be simplified to make the causal structure of the evolution apparent

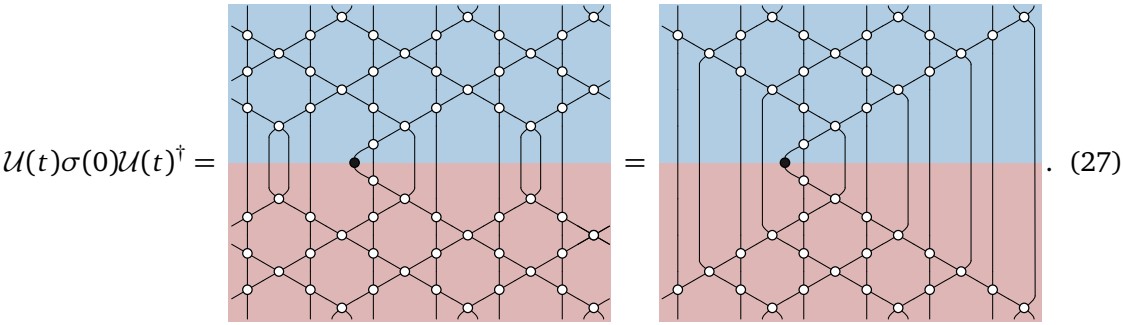

$$\mathcal{U}(t)\sigma(0)\mathcal{U}(t)^{\dagger} = \qquad\qquad = \qquad\qquad . \qquad (27)$$

The dynamical correlation function is defined as $C(x,y,t) = \mathrm{tr}[\sigma(x,t)\rho(y,0)]/\mathrm{tr}[\mathbb{1}]$. The time-evolved operator originally initialized at site $x$ is contracted with a local operator $\rho(y,0)$ at site $y$, and at the end the trace is taken. Because of the horizontal unitarity of the biunitary connections, for the current choice of initial sublattice, the correlation function is only non-vanishing if $\rho$ is placed on the right edge of the light-cone

$$C(x,y,t) = \qquad\qquad\qquad . \tag{28}$$

This result directly follows the observation that, in the absence of an operator $\rho$ on the right edge of the light cone, horizontal unitarity can be used to fully contract this outer edge, leading to a vanishing correlation function (similar to the 'telescoping' argument used in dual-unitary circuits [2]). Further using horizontal unitarity, this expression simplifies to

$$C(x,y,t) = \qquad\qquad\qquad . \tag{29}$$

This expression is efficiently computable as it can be expressed through a low-dimensional quantum channel, again in the same way as for dual-unitary circuits [3,4].

A similar computation is carried out when placing the initial operator one site further to the right. Using vertical unitarity the time-evolved operator can first be simplified to

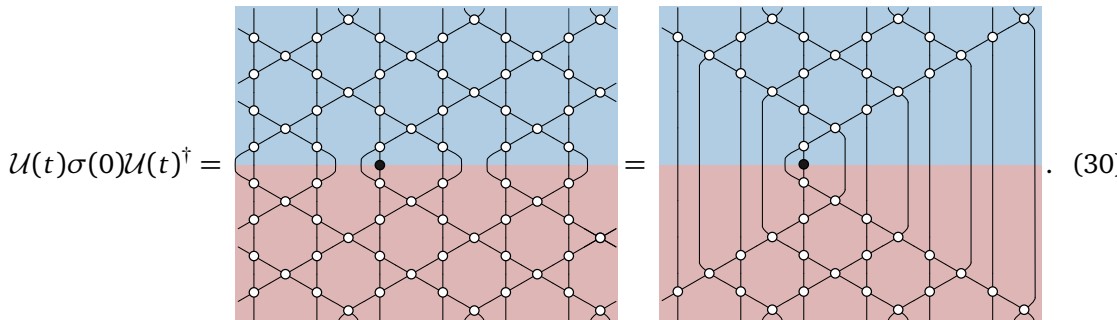

$$\mathcal{U}(t)\sigma(0)\mathcal{U}(t)^{\dagger} = \qquad = \qquad . \tag{30}$$

The only nonvanishing correlator is now obtained for $x = y$, leading to a diagram of the form

$$C(x,x,t) = \quad 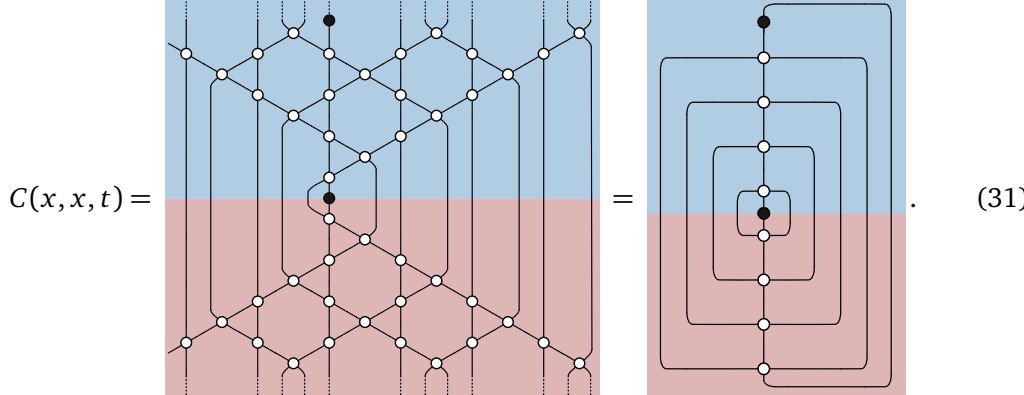 \quad . \tag{31}$$

For the two other choices of initial lattice sites the procedure is analogous to the above. We hence recover the distinct rays in spacetime with $x = \pm t$ and $x = 0$.

## 3.2 Thermalization

In this section we investigate the thermalization of finite subsystems. We consider quenches from a particular class of initial states satisfying a spatial unitarity condition, so-called solvable states [8,39]. We show, that for any shading pattern, finite contiguous subsystems relax to the maximally mixed state exactly after a finite number of steps. This result generalizes previous results on exact thermalization in (generalized) dual-unitary circuits [8,33,36,39].

Solvable states were introduced in Ref. [8] and generalized to biunitary connections in Ref. [39]. We follow the exposition in Ref. [39]. A *solvable tensor* is defined as a vertex

$$\tag{32}$$

which satisfies a horizontal unitarity condition

$$\tag{33}$$

A *solvable state* is then constructed by contracting solvable tensors horizontally in analogy to matrix product states.

After a quench, the density matrix of a finite contiguous subregion $A$ of length $|A|$ reads

$$\rho_A(t) = \text{tr}_{\bar{A}}\big[\mathcal{U}(t)|\psi\rangle\langle\psi|\mathcal{U}^\dagger(t)\big] = \quad 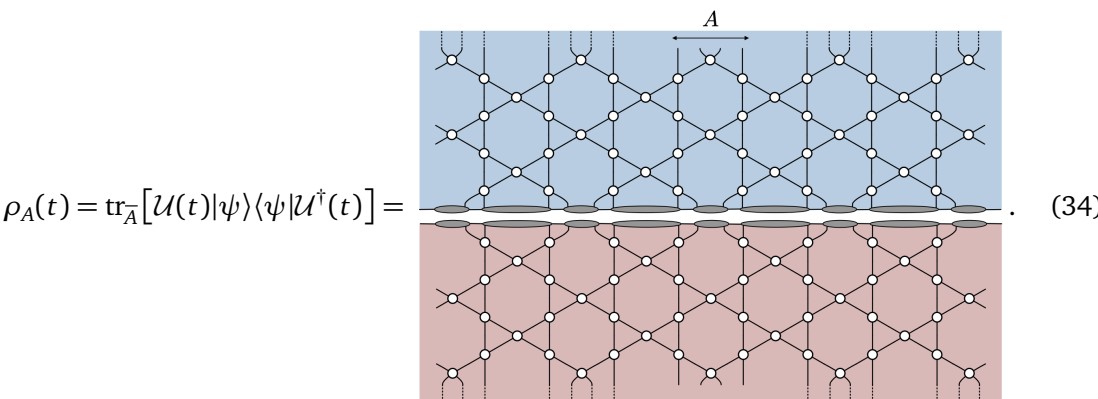 \quad . \tag{34}$$

Using vertical unitarity, this expression can be simplified to yield

$$\rho_A(t) = \quad \text{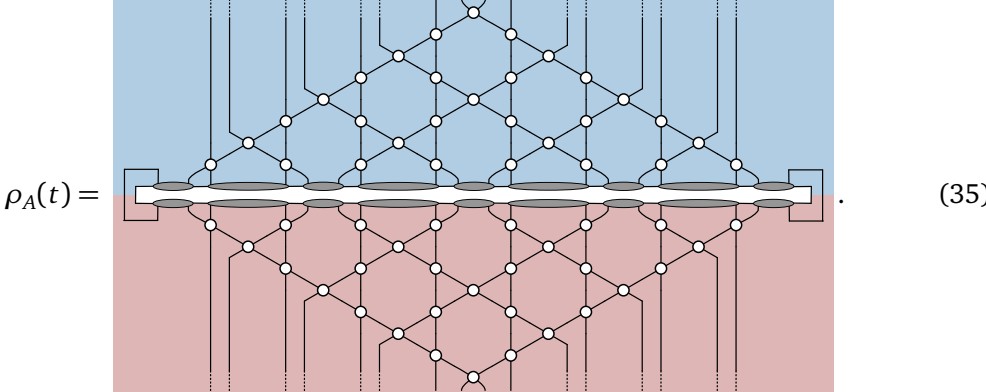} \quad . \tag{35}$$

Further using the defining condition of solvable states and horizontal unitarity gives

$$\rho_A(t) = \quad \text{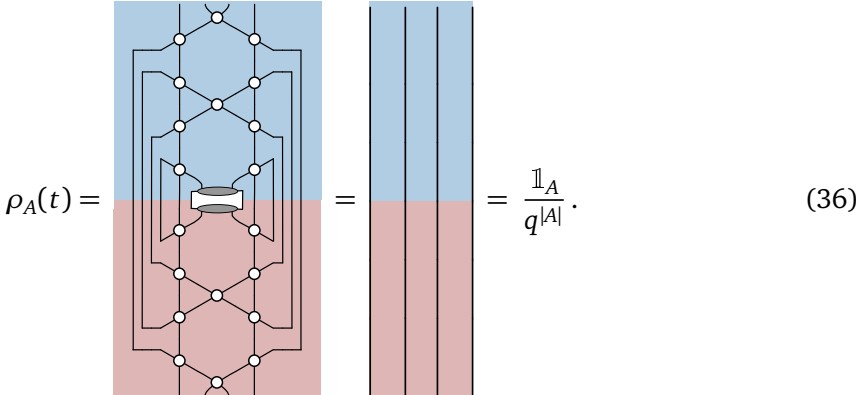} \quad = \quad = \frac{\mathbb{1}_A}{q^{|A|}} . \tag{36}$$

The result is a maximally mixed state, which is the expected thermal state for a system without conservation laws. For a subsystem consisting of $2\ell$ sites, exact thermalization happens after the application of $t = \ell$ layers. Note that the specific dimension of the Hilbert space and number of time steps will generally depend on the choice of shading pattern.

## 3.3 Generalized dual-unitarity and triunitarity in the kagome lattice

We can now consider different realizations of the biunitary kagome lattice, which are all guaranteed to exhibit the same dynamics of correlation functions. Periodic shading patterns result in lattices in which the underlying geometry is directly made apparent. When all faces remain unshaded, the resulting circuit consists of a kagome lattice with dual-unitary gates at the vertices:

$$\text{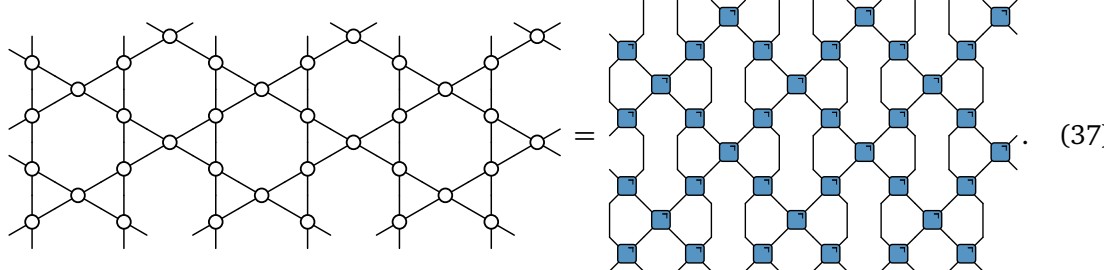} \tag{37}$$

This model can be thought of as a triunitary circuit by identifying right-pointing triangles with three-site gates [31, 66].

When all triangles are shaded, but the hexagons remain unshaded, each vertex borders two shaded areas that are not adjacent. The circuit can therefore be expressed in terms of complex Hadamard matrices. In this case these matrices are placed on the links of a honeycomb lattice

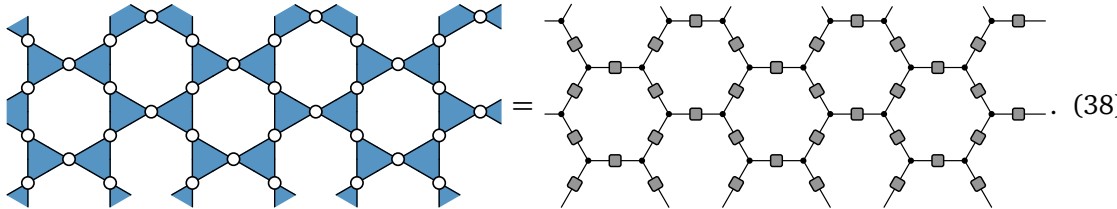

$$ \tag{38} $$

This construction previously appeared in Refs. [35–37]. In the quantum circuit representation the shaded regions result in delta tensors, which we graphically define as

$$ \overset{a}{\underset{b}{\diagdown}}\!\!\bullet\!\!-c = \delta_{abc} \,. \tag{39} $$

Because of the biunitarity of the complex Hadamard matrices, the vertical gates can be interpreted as single-site unitary gates and the horizontal gates correspond to two-site controlled phase gates, where the delta tensors enforce that these are diagonal.

Similarly, shading all hexagons, but not the triangles, yields complex Hadamard matrices on the links of a triangular lattice, a construction that appeared in Ref. [35],

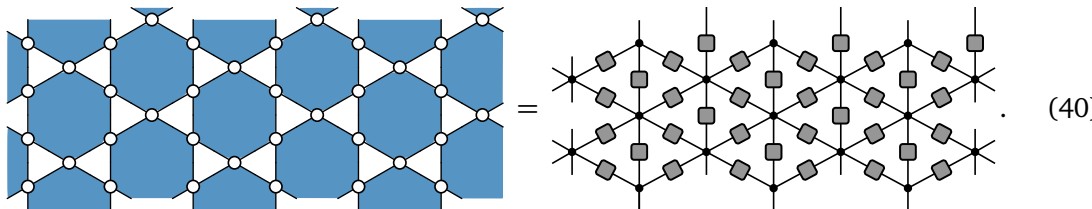

$$ \tag{40} $$

The black circles again represent delta tensors, now with six indices. Novel solvable circuit models can be straightforwardly obtained by considering different shading patterns. For a fully shaded kagome lattice, we recover a triunitary "interactions round-a-face" circuit with the DU2 property. Expressed in terms of quantum crosses, the circuit takes the form of a honeycomb lattice with the crosses placed on the links interacting with crosses situated on the same hexagon

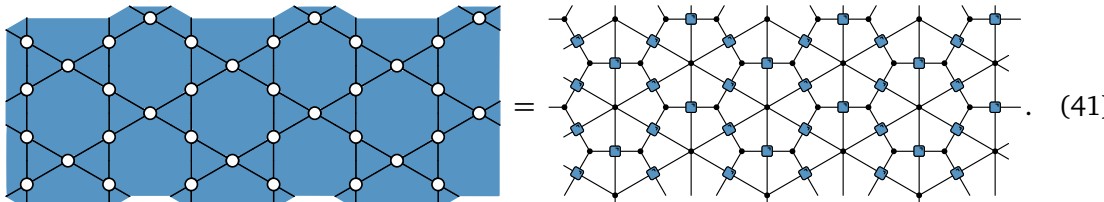

$$ \tag{41} $$

We can also consider a periodic pattern where diamonds of biunitaries are shaded, returning a lattice of quantum Latin squares connected by complex Hadamard matrices:

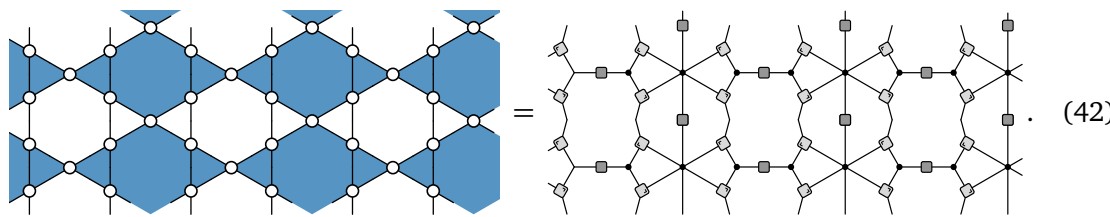

$$ \tag{42} $$

As one final example, consider a lattice where columns are alternately shaded:

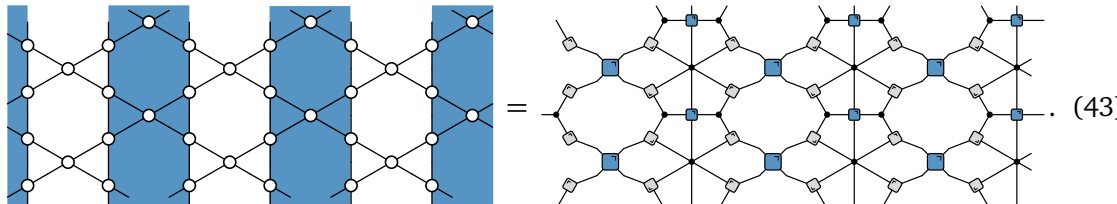

$$\tag{43}$$

This shading returns a regular pattern of columns consisting of quantum crosses and quantum Latin squares connected by dual-unitary gates.

We emphasize that while all these models exhibit the same light-cone correlation dynamics of triunitary and hierarchical DU2 circuits, the corresponding algebraic manipulations are vastly different – it is only their expression in terms of the shaded calculus that highlights the similarities. In all these models, the connection with triunitarity and hierarchical dual-unitarity can be made more explicit by observing that two different choices of unit cell can be identified in the kagome lattice. These are made explicit in the biunitary kagome lattice in the figure below:

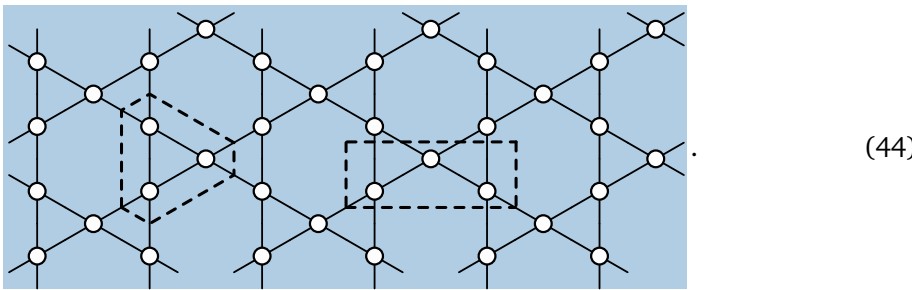

$$\tag{44}$$

Crucially, the triangular choice of unit cell satisfies a triunitary condition, whereas the rectangular choice of unit cell satisfies the hierarchical DU2 condition. The full lattice is obtained by arranging these unit cells according to the triunitary structure [Eq. (4)] and brickwork structure [Eq. (2)] respectively.

**Hierarchical DU2 condition.** The circuit on the kagome lattice can be generated from the following building block

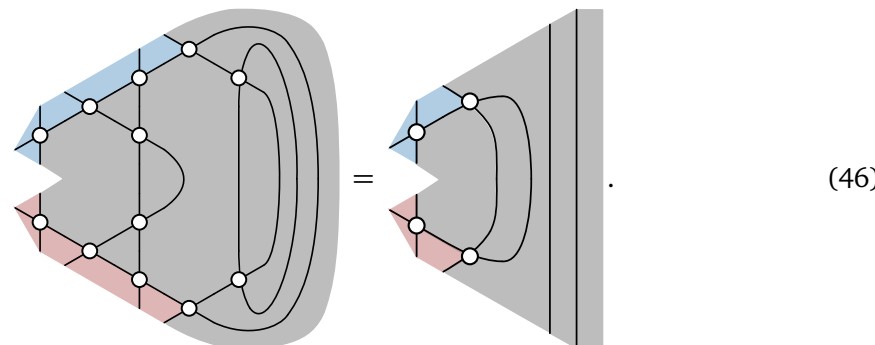

$$\tag{45}$$

by arranging it in a brickwork manner. Although this object does not necessarily represent a two-site unitary gate – it might carry more than four indices – it satisfies a condition generalizing the DU2 condition. Specifically, it follows from biunitarity that

$$\tag{46}$$

Here, any area can be either shaded or unshaded, provided overall consistency. Each biunitary connection in this diagram is related to its mirrored counterpart by Hermitian conjugation. This equation is analogous to the DU2 condition (6) in that the left-hand side involves a contraction of two building blocks and their conjugates that can be reduced to identities for the indices carried by the outer building block.

**Triunitary condition.**    Alternatively to the above point of view, the kagome lattice construction can also be seen as a generalized version of a triunitary circuit. For this, we note that the circuit can also be generated by the following building block

$$\text{(47)}$$

For any shading, this building block satisfies a set of equations that can be interpreted as a generalized form of triunitarity (5). In addition to the usual unitarity condition, we have

$$\text{(48)}$$

## 3.4   Example constructions

The power of the approach via biunitary connections is that it provides a common language for models that seem superficially very different. It enables the derivation of general physical properties that are shared by all models that can be expressed as biunitary connections, even when these models differ in their Hilbert spaces, algebraic properties, and in the geometry of connections and corresponding interactions in spacetime (as exemplified by the examples above). In the following, we present a survey of models which are obtained from the simplest periodic shading patterns of the kagome lattice. Because of their periodicity, these examples give rise to a variety of DU2 and triunitary gate constructions, both novel and already known. While the general features of the resulting dynamics is very similar in all cases, different shadings are suitable for constructing circuits with certain special properties. For instance, the unshaded lattice turns out to enable the construction of non-ergodic circuits in a simple and transparent manner. Other shadings might be better suitable for implementation on quantum devices, such as the bipartite shading patterns leading to arrangements of complex Hadamard matrices, which can be implemented using only one-qubit operations and phase gates.

### 3.4.1   DU2 and triunitary gates from dual-unitary gates

One possibility is to leave all areas unshaded. The circuit then consists of a kagome lattice with dual-unitary gates (of dimension $q$) at the vertices. We can identify a gate acting on two

qudits of dimension $q^2$, which satisfies the DU2 condition

$$\tag{49}$$

The $q^2$-dimensional Hilbert space consists of two $q$-dimensional Hilbert spaces grouped together. This result can be seen summarized in noting that three DU gates of dimension $q$ can always be used to form a DU2 gate of dimension $q^2$. Such circuits were also discussed in Ref. [66] as examples of triunitary circuits, where a triunitary three-site gate can similarly be composed out of three dual-unitary gates:

$$\tag{50}$$

However, the above shows that they can also be seen as DU2 circuits, providing an example were both notions coincide.

The DU2 property enables us to characterize the entanglement dynamics of this circuit. Knowledge of the Schmidt rank $\mathcal{R}$ and the Hilbert space dimension $d$ is sufficient to compute the entanglement velocity according to Eq. (8). It follows directly from the construction that $\mathcal{R} = d = q^2$, so $v_E = \log \mathcal{R} / \log d^2 = 1/2$. More generally, the DU2 gate can also be constructed from underlying DU gates whose Hilbert space dimensions are not the same. The most general case is obtained by taking a central gate acting on $\mathbb{C}^{q_2} \otimes \mathbb{C}^{q_2}$ and left and right gates acting on $\mathbb{C}^{q_2} \otimes \mathbb{C}^{q_1}$ and $\mathbb{C}^{q_1} \otimes \mathbb{C}^{q_2}$ respectively. This yields a DU2 gate acting on $\mathbb{C}^{q_1 q_2} \otimes \mathbb{C}^{q_1 q_2}$. The entanglement velocity of the resulting circuit can again be read off from the Schmidt rank $\mathcal{R} = q_2^2$ and $d = q_1 q_2$ to be

$$v_E = \frac{\log q_2^2}{\log q_1^2 + \log q_2^2}. \tag{51}$$

The construction furthermore enables us to construct non-ergodic DU2 circuits possessing solitons. Consider a dual-unitary gate $U$ which possesses a bidirectional soliton $\sigma$, i.e. a one-site operator that is translated by one site under the action of the unitary (see Ref. [67] for a discussion of solitons in dual-unitary circuits). We assume that $\sigma$ satisfies

$$U(\sigma \otimes \mathbb{1})U^\dagger = \mathbb{1} \otimes \sigma, \quad U(\mathbb{1} \otimes \sigma)U^\dagger = \sigma \otimes \mathbb{1}. \tag{52}$$

Then, the DU2 gate $U_K$ constructed from three copies of $U$ satisfies

$$U_K((\sigma \otimes \mathbb{1}_q) \otimes \mathbb{1}_{q^2})U_K^\dagger = \mathbb{1}_{q^2} \otimes (\sigma \otimes \mathbb{1}_q), \tag{53a}$$

$$U_K((\mathbb{1}_q \otimes \sigma) \otimes \mathbb{1}_{q^2})U_K^\dagger = (\sigma \otimes \mathbb{1}_q) \otimes \mathbb{1}_{q^2}, \tag{53b}$$

$$U_K(\mathbb{1}_{q^2} \otimes (\sigma \otimes \mathbb{1}_q))U_K^\dagger = \mathbb{1}_{q^2} \otimes (\mathbb{1}_q \otimes \sigma), \tag{53c}$$

$$U_K(\mathbb{1}_{q^2} \otimes (\mathbb{1}_q \otimes \sigma))U_K^\dagger = (\mathbb{1}_q \otimes \sigma) \otimes \mathbb{1}_{q^2}. \tag{53d}$$

The first (fourth) relation implies the existence of a soliton $\sigma \otimes \mathbb{1}_q$ ($\mathbb{1}_q \otimes \sigma$) moving along the edge of the light cone to the right (left). The second and third relation lead to solitons which do not move after application of a full circuit layer. In total, there are one right-moving, one left-moving, and two localized solitons per unit cell. Note that the existence of such solitons directly implies non-ergodicity and leads to super-integrable dynamics if the set of solitons is complete [56, 67, 68].

### 3.4.2 DU2 gate from complex Hadamard matrices

There are two inequivalent periodic bipartite shadings of the kagome lattice, returning the honeycomb and triangular lattices, respectively. The resulting circuits (38) and (40) can be expressed solely in terms of complex Hadamard matrices. We first consider the partially shaded case where

$$
\tag{54}
$$

This shading pattern now defines a two-site unitary gate as the building block of the honey-comb lattice. This construction recovers a parametrization of DU2 gates in terms of complex Hadamard matrices equivalent to the one presented in Ref. [35]:

$$
\tag{55}
$$

The hierarchical condition Eq. (46) for this gate now reduces to the usual DU2 conditions for two-site gates:

$$
\tag{56}
$$

It has Schmidt rank $\mathcal{R} = q$ and dimension $q$, leading to an entanglement velocity $v_E = 1/2$.

For the case of the triangular lattice, it is more convenient to consider the building block

$$
\tag{57}
$$

This object now represents a controlled two-site gate. Remarkably, for this unit cell, the generalized triunitary condition (48) now returns the DU2 condition on the two-site gate. The two nontrivial conditions read

$$
\tag{58}
$$

Here, the red hue indicates a Hermitian conjugate. The DU2 conditions follow from these.

For the first condition, the first equality in Eq. (58) can be applied twice to yield

$$ \tag{59} $$

The second condition is reduced with the help of the second equality in Eq. (58), resulting in

$$ \tag{60} $$

This gate hence satisfies the DU2 condition and also leads to $\nu_E = 1/2$.

### 3.4.3 DU2 gate from unitary error bases

Another possible pattern is formed by shading only every right-pointing triangle. This yields a gate of dimension $q^2$ composed of three UEBs

$$ \tag{61} $$

To view this construction as a two-site gate, the outgoing legs of the UEBs are grouped together to form a Hilbert space of dimension $q^2$. The other site is given by the control leg of the UEB which is also of dimension $q^2$. The Schmidt rank of this gate is $\mathcal{R} = d = q^2$, which again implies $\nu_E = 1/2$. As previously, the circuit can alternatively be viewed through the lens of triunitarity. The alternative unit cell returns a new construction of triunitary gates

$$ \tag{62} $$

### 3.4.4 Constructions from quantum Latin squares

When the shading pattern has a larger unit cell, it may be necessary to consider a larger building block to identify a DU2 gate. In the case of the circuit (42), this is achieved by

doubling the unit cell. The object

$$
\text{(63)}
$$

satisfies the DU2 condition, as can be checked by a tedious calculation. The circuit (43) gives rise to an analogous construction.

As a remark, quantum Latin squares also naturally constitute DU2 gates. A QLS can be thought of as a controlled gate

$$
U_{ab,cd} = \delta_{ac}(Q_{ab})_d = \quad \text{(64)}
$$

Using the defining properties of QLS it can be shown that this gate satisfies one of the two DU2 conditions

$$
\text{(65)}
$$

In the first step we have used the completeness of columns, and again in the second step. This condition results in solvability of the dynamics in the spacetime region $x \geq 0$. To fulfill the second condition, the QLS however has to satisfy an additional requirement: The reshaped object $(\tilde{Q}_{ab})_c = (Q_{ac})_b$ has to satisfy the QLS property as well.

### 3.4.5  Triunitary interactions round-a-face

Before moving on to analyze the "interactions round-a-face" circuit (41), we first introduce the notion of *triunitary interactions round-a-face*, or *triunitary quantum crosses*. We then show that Eq. (41) constitutes an example of a triunitary interactions round-a-face circuit.

Consider a set of $q^2 \times q^2$ unitary matrices

$$
(G_{ad})_{bc,ef} = \quad \text{(66)}
$$

with every index running from 1 to $q$. This can be thought of as a double-controlled two-site gate. We call these gates triunitary interactions round-a-face, if the matrices $(G_{bf})_{cd,ae}$ and $(G_{ec})_{ab,fd}$ are also unitary. Graphically, the unitarity of $G_{ad}$ is expressed as

$$
\text{(67)}
$$

and the other two conditions as

$$
\quad (68)
$$

These two conditions express unitarity when the time direction is rotated by $\pm\pi/3$. This definition naturally generalizes dual-unitary interactions round-a-face to triunitarity. When arranged on a triangular lattice these gates define a unitary circuit with properties analogous to triunitary circuits:

$$
\quad (69)
$$

The fully shaded kagome lattice (41) realizes such a triunitary interactions-round-a-face circuit. Identifying

$$
\quad (70)
$$

it can be seen that the generalized triunitarity condition implies Eq. (68).

Finally, we discuss the entanglement velocity in triunitary interactions round-a-face circuits. Since there is no description in terms of two-site gates fulfilling the DU2 condition (not even when considering a larger unit cell), the entanglement velocity cannot simply be read off from the Schmidt rank. Moreover, the Schmidt rank is ill-defined in this case. For this reason, we explicitly show that $v_E = 1/2$ for quenches from solvable states in triunitary interactions round-a-face, in accordance with previous examples of shaded kagome circuits.

For quantum crosses, solvable tensors are given by a set of $q$ $q \times q$ unitary matrices

$$
\quad (71)
$$

We consider an infinite chain and compute the $n$th Rényi entropy of the time-evolved state with respect to a cut bisecting the chain as

$$
S_A^{(n)}(t) = \frac{1}{1-n} \log \operatorname{tr}_{\bar{A}}[\rho_A(t)^n] \,, \quad (72)
$$

where $\rho_A(t)$ is the reduced density matrix on the half-chain $A$. This half-chain entropy captures the early-time entanglement growth of contiguous subsystems in the scaling limit. To express the Rényi entropies graphically, we briefly introduce some standard notation [41]. We work in the $2n$-times replicated space $(\mathcal{H} \otimes \mathcal{H}^*)^{\otimes n}$, where $\mathcal{H}$ denotes the Hilbert space of a single chain.

Index contractions between the replicas can be seen as vectors in this space. We introduce the following two contractions

$$\overset{2n}{\overbrace{\quad}} \qquad \overset{2n}{\overbrace{\quad}}$$

$$\mathbin{\vert}_\circ = \overbrace{\mathsf{U} \cdots \mathsf{U}}, \qquad \mathbin{\vert}_\square = \overbrace{\mathsf{U} \cdots \mathsf{U}}. \tag{73}$$

These vectors have norm $q^{n/2}$ and their overlap is $\square\!\!-\!\!\circ = q$. With these vectors, the horizontal unitarity of the solvable tensor can be written as

$$\begin{array}{c}\circ\\ \square\!\!-\!\!\blacksquare\!\!-\!\!\circ\\ \circ\end{array} = \text{---}\circ, \tag{74}$$

where the diagram now represents $2n$ copies of the state and its Hermitian conjugate. Analogous expressions hold when contracting from the other side and when contracting with the other index contraction pattern. We now graphically represent $\mathrm{tr}_{\bar{A}}[\rho_A(t)^n]$ as

$$\mathrm{tr}_{\bar{A}}[\rho_A(t)^n] = \dots \qquad \dots , \tag{75}$$

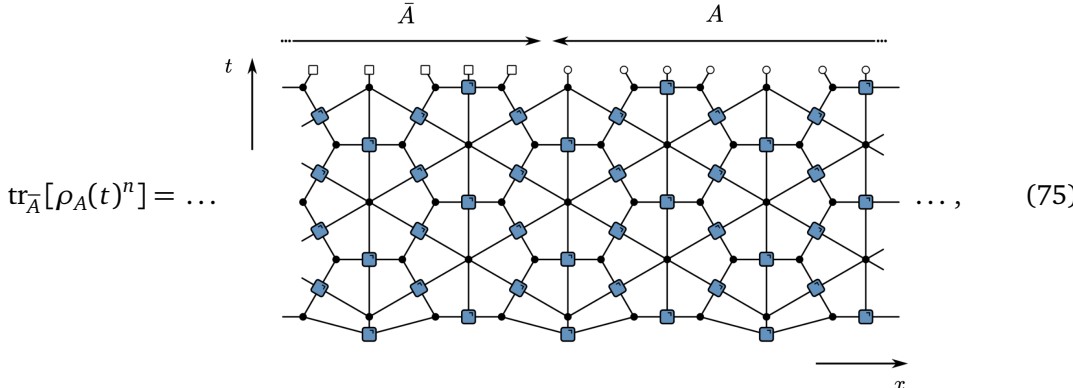

where we have used the same notation for the replicated quantum crosses and solvable tensors as for the unreplicated ones. Using vertical unitarity of the replicated quantum crosses, we find

$$\mathrm{tr}_{\bar{A}}[\rho_A(t)^n] = \qquad . \tag{76}$$

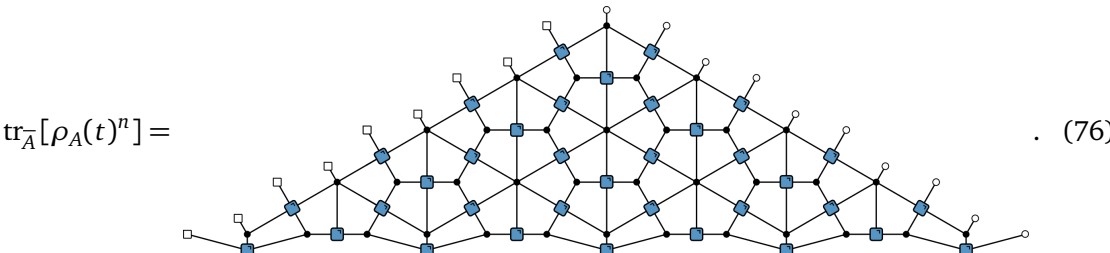

This diagram can be further simplified using the other unitary directions to read

$$\mathrm{tr}_{\bar{A}}[\rho_A(t)^n] \propto \qquad . \tag{77}$$

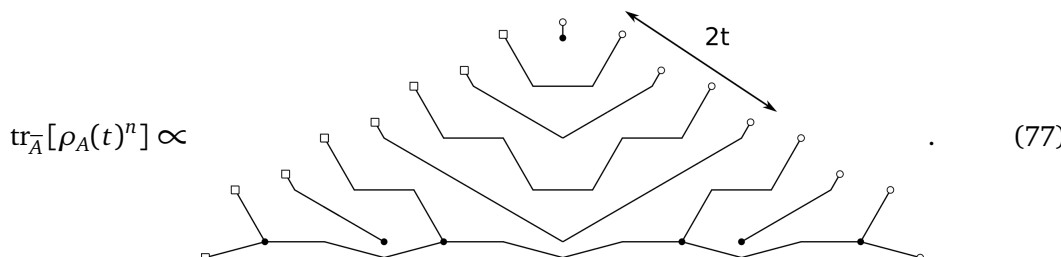

This final diagram can now be interpreted as representing $2t + 1$ Bell pairs connecting $A$ to its complement. The open contractions and the contractions that end on the bottom Bell pair exactly cancel with the normalization of the solvable state, such that we obtain

$$S_A^{(n)}(t) = (2t + 1)\log(q). \tag{78}$$

Since all Rényi entropies are equal, this also extends to the von Neumann entropy. This result corresponds to an entanglement velocity of $v_E = 1/2$, since only $2t + 1$ of the $4t + 1$ causally connected sites are connected by a Bell pair.

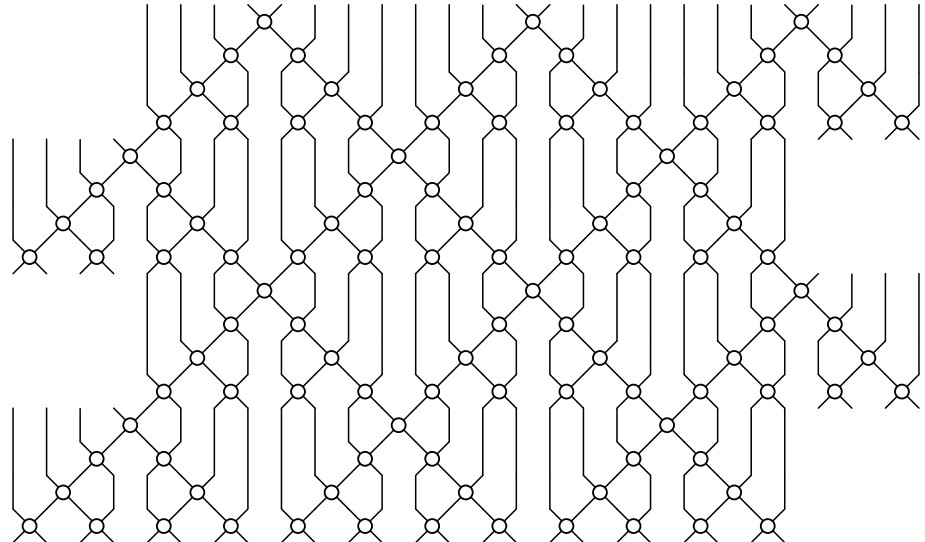

Figure 1: Patch of a DU2 circuit constructed from the nested kagome gate $U_+$.

## 3.5 Nested kagome lattice

The kagome lattice construction has the limitation that it constrains the possible entanglement velocities to those of Eq. (51). This restriction can be seen as a consequence of the presence of a six-fold symmetry in spacetime. Equivalently, it can be seen as a consequence of generalized triunitarity. For generality, it is however desirable to obtain constructions of DU2 circuits with different entanglement velocities. In the following we show how a nesting transformation of the kagome lattice [see Fig. 1] gives rise to entanglement velocities $v_E = 2^{-n}$ with $n \in \mathbb{N}$. (We discuss how to obtain more general values of $v_E$ in the next section.)

For which $q$-dimensional gates does the construction of Eq. (49) yield a $q^2$-dimensional DU2 gate? We have shown above that it is sufficient to choose DU gates. However, it is easy to see that choosing a $q$-dimensional DU2 gate $U_{\text{DU2}}$ and composing it according to Eq. (49) also yields a $q^2$ DU2 gate. This observation enables the nesting of different constructions of DU2 gates. For instance, nesting the construction (49) with itself yields a DU2 gate of the form

$$U_+ = \qquad , \tag{79}$$

where each vertex represents a dual-unitary gate. Alternatively, it is possible to nest (49) with its transpose

$$U_- = \qquad . \tag{80}$$

Both $U_+$ and $U_-$ can be seen as acting on two qudits of dimension $d = q^4$. Their Schmidt rank is $\mathcal{R}_\pm = q^2$, yielding an entanglement velocity $v_E = 1/4$. Here the Schmidt rank can be read off from the single dual-unitary matrix connecting both halves of the construction.

This procedure can be repeated an arbitrary number of times, each time choosing either to nest Eq. (49) or its transpose on the next level. For such an $n$-times nested gate, the Hilbert

space dimension of the qudits is $q^{2^{n+1}}$ and the Schmidt rank is $\mathcal{R} = q$ since both halves of the construction remain connected by a single dual-unitary gate, implying $\nu_E = 1/2^{n+1}$. Interestingly, while the nesting transformation preserves DU2 it breaks triunitarity, since the three-site gate formed according to Eq. (50) is no longer triunitary. This is unsurprising from the point of view of entanglement dynamics: triunitarity fixes the entanglement velocity to be $\nu_E = 1/2$ [31] (in our units).

Note that the circuits constructed through this nesting procedure will again exhibit (some degree of) solvability when shaded regions are introduced. We however postpone a discussion of this solvability to future works.

# 4 Multilayer constructions

The conjecture of Ref. [35] implies that the Schmidt ranks that DU2 circuits can have are strongly restricted, in turn restricting the possible entanglement velocities beyond the quantization condition (8). For a DU2 gate in Hilbert space dimension $q$ with prime factorization $q = \prod_{i=1}^{m} q_i$, the conjecture is that the possible values of $\mathcal{R}$ are

$$\mathcal{R} = \prod_{i=1}^{m} q_i^{\nu_i}, \quad \nu_i \in \{0, 1, 2\}, \tag{81}$$

meaning that each prime factor may only contribute a Schmidt rank of $\{1, q_i, q_i^2\}$. This implies that the possible entanglement velocities are given by

$$\nu_E = \frac{\sum_{i=1}^{m} \log\left(q_i^{\nu_i}\right)}{\log q^2}, \quad \nu_i \in \{0, 1, 2\}. \tag{82}$$

In particular, when $q$ is prime entangling DU2 gates have $\nu_E = 1/2$ ($\nu_E = 0$ is non-entangling and $\nu_E = 1$ implies DU). Therefore, to find DU2 gates with $\nu_E \neq 1/2$ one must turn to composite Hilbert space dimensions. A simple way to construct DU2 circuits in composite dimensions with an entanglement velocity of the form (82) is by forming tensor products of DU2 gates with dual-unitary gates and product gates, as described in Ref. [35]. In this construction, a tensor product structure is imposed on the local Hilbert space of the chain, and the factors are evolved separately by gates coupling only like factors of neighboring sites. In the quantum computing literature such gates are called *transversal*. Both dual-unitary gates and product gates satisfy the DU2 condition, and the former are maximally entangled whereas the latter are unentangled. The resulting tensor product gates will trivially satisfy the hierarchical DU2 condition since all constituting gates satisfy this condition, with the entangling properties following directly from those of the underlying gates.

While such models are convenient testbeds to explore the possible properties of DU2 gates, all their properties can be understood purely based on the behavior of the individual layers. From a geometric perspective such a tensor product construction can be understood as a multilayer system of circuits superimposed on each other with no coupling between the layers. In this section, we introduce multilayer circuits of biunitary connections where the layers are genuinely coupled to each other. This is natural in the language of biunitary connections, as multilayer biunitary connections arise naturally in the study of quantum combinatorial data and many constructions of CHM, UEB, and QLS can be understood through this lens [50]. By coupling layers of different lattice structure together, we are able to go beyond circuits with either square or hexagonal symmetry enabling us to obtain models with the full range of entanglement velocities (82). We also show how multilayer constructions can be used to systematically construct instances of DU3 circuits.

## 4.1 Multilayer biunitary connections

In this section, we give a short introduction to multilayer biunitary connections, following Ref. [50]. Multilayer biunitary connections are biunitary connections that cannot be drawn in a plane without overlapping legs or shaded areas. Compared to the instances of biunitary connections that we have seen so far, these objects carry additional indices that are associated to legs or shaded areas that are drawn in layers above each other. The simplest examples are given by controlled families of one-layer biunitary connections. The control index is associated to a shaded half-plane that is drawn as an additional layer.

A controlled family of complex Hadamard matrices can e.g. be drawn as

$$H^c_{ab} = \quad . \tag{83}$$

The range of the control index $c$ is arbitrary. The horizontal and vertical unitarity conditions merely enforce that $H^c$ is a complex Hadamard matrix for fixed $c$. Note that the red color here does not indicate the Hermitian conjugate but rather the shaded half-plane.

Multilayer biunitary connections have proven useful in the construction of quantum combinatorial data, and various well known constructions of complex Hadamard matrices can be formulated in terms of multilayer biunitary connections. The Hosoya-Suzuki (HS) construction [69] gives a dimension-$dq$ complex Hadamard matrix from $q$ dimension-$d$ and $d$ dimension-$q$ Hadamard matrices

$$H_{ab,cd} = J^b_{ac} K^c_{bd} . \tag{84}$$

This can be graphically represented as

$$H_{ab,cd} = \quad . \tag{85}$$

Related to this is Diţă's construction [70] taking one dimension-$d$ and $d$ dimension-$q$ complex Hadamard matrices as input and also yielding a dimension-$dq$ complex Hadamard matrix

$$H_{ab,cd} = J_{ac} K^c_{bd} = \quad . \tag{86}$$

For convenience, we introduce the following graphical notation to refer to any of the two above constructions

$$H = \quad . \tag{87}$$

Additionally, we also take this notation to encompass complex Hadamard matrices which take a tensor product form $H_{ab,cd} = J_{ac}K_{bd}$. In this case the two layers decouple, while for the HS and Diţă constructions, the layer coupling is mediated by the control indices.

For concreteness, consider the two-dimensional complex Hadamard matrices

$$H_0 = \begin{pmatrix} 1 & 1 \\ 1 & -1 \end{pmatrix}, \quad H_1 = \begin{pmatrix} e^{i\phi} & 1 \\ 1 & -e^{-i\phi} \end{pmatrix},$$  (88)

where $\phi$ is a real parameter. A four-dimensional CHM can be formed by taking a tensor product

$$H_{\mathrm{TP}} = H_0 \otimes H_1 = \begin{pmatrix} e^{i\phi} & 1 & e^{i\phi} & 1 \\ 1 & -e^{-i\phi} & 1 & -e^{-i\phi} \\ e^{i\phi} & 1 & -e^{i\phi} & -1 \\ 1 & -e^{-i\phi} & -1 & e^{-i\phi} \end{pmatrix}.$$  (89)

The HS and Diţă constructions enable finding four-dimensional entangling complex Hadamard matrices. Defining the family $J = \{H_0, H_1\}$ we find

$$H_{HS} = \sum_{a,b,c,d} \langle a|J^b|c\rangle \langle b|J^c|d\rangle |ab\rangle\langle cd| = \begin{pmatrix} 1 & 1 & e^{i\phi} & 1 \\ e^{i\phi} & -e^{i\phi} & 1 & -e^{-i\phi} \\ 1 & 1 & -e^{i\phi} & -1 \\ 1 & -1 & -e^{-i\phi} & e^{-2i\phi} \end{pmatrix},$$  (90)

and from Diţă's construction

$$H_D = \sum_{a,b,c,d} \langle a|H_0|c\rangle \langle b|J^c|d\rangle |ab\rangle\langle cd| = \begin{pmatrix} 1 & 1 & e^{i\phi} & 1 \\ 1 & -1 & 1 & -e^{-i\phi} \\ 1 & 1 & -e^{i\phi} & -1 \\ 1 & -1 & -1 & e^{-i\phi} \end{pmatrix}.$$  (91)

These constructions can be iterated arbitrarily often. Similar construction schemes can be devised for other quantum combinatorial data, such as UEB or QLS [50].

## 4.2 Multilayer DU circuits

In this section we illustrate the idea of multilayer biunitary circuits by considering dual-unitary circuits constructed from multilayer biunitary connections. As in this case the lattice structure of the layers is identical, the entangling properties coincide with those of known constructions of dual-unitary circuits. We first give examples of constructions of DU gates from lower-dimensional objects.

The simplest manner in which a DU gate can be constructed from biunitary connections is from four complex Hadamard matrices [57]

$$U = \vcenter{\hbox{[diagram]}} .$$  (92)

Written out in matrix elements, this returns $U_{ab,cd} = K_{ab}K_{bd}K_{dc}K_{ca}/q$, where $K$ denotes four (possibly different) $q$-dimensional complex Hadamard matrices. A multilayer gate is obtained if these matrices are of the form (87), such as when they are obtained from the Hosoya-Suzuki

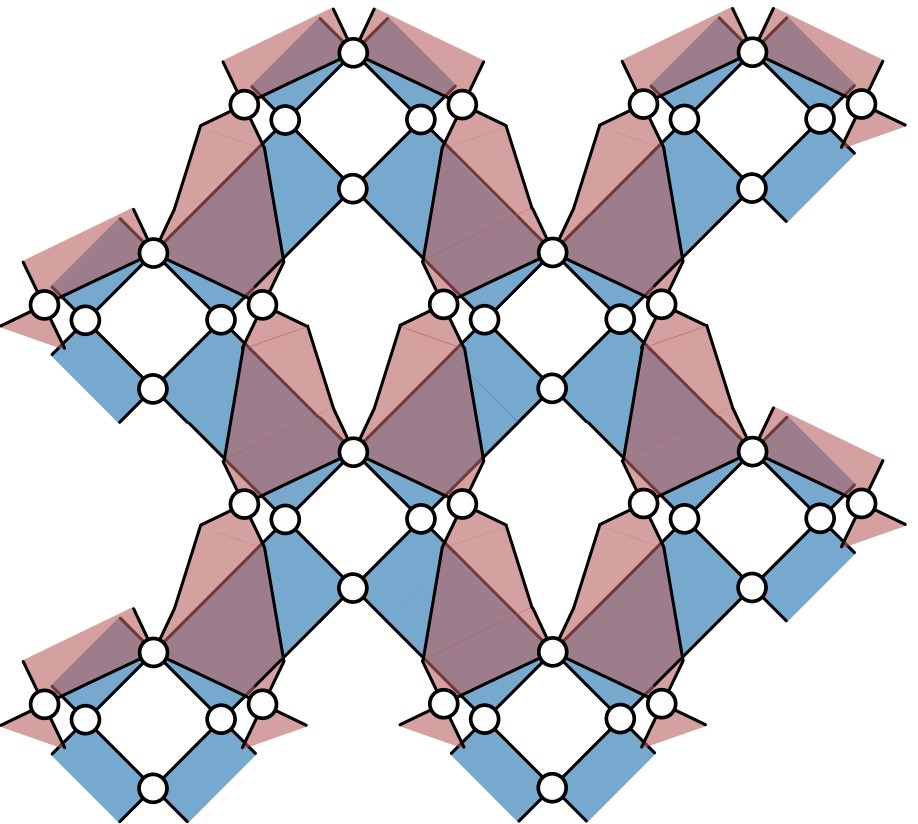

Figure 2: Multilayer circuit coupling a DU circuit of square lattice geometry (blue) to a kagome lattice DU2 circuit (red). This yields an entanglement velocity intermediate between $v_E = 1$ (DU) and $v_E = 1/2$ (DU2).

or Diţă's construction. The multilayer complex Hadamard matrix act as couplings between the layers and the amount of coupling can be adjusted by choosing some of the multilayer matrices to be of tensor product form, thus not contributing to the coupling. By increasing the number of HS constructed complex Hadamard matrices, the coupling between the layers is increased as well. The general form of a multilayer DU gate constructed from multilayer complex Hadamard matrices is

$$ U = \quad \text{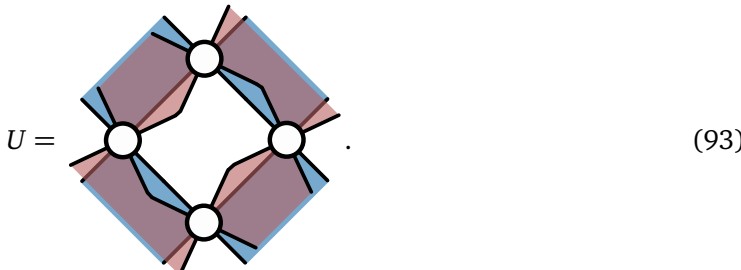} \quad . \tag{93} $$

As any multilayer complex Hadamard matrices satisfies the usual CHM properties, these constructed gates are dual-unitary. Using $H$ to denote (possibly different) multilayer $q_1 q_2$-dimensional CHMs with matrix elements $H_{ab,cd}$, following Eqs. (85) and (86), the resulting unitary has matrix elements

$$ U_{\mathbf{ab},\mathbf{cd}} = H_{a'a,b'b} H_{b'b,d'd} H_{d'd,c'c} H_{c'c,a'a} / (q_1 q_2), \tag{94} $$

where indices in the composite Hilbert space are denoted by $\mathbf{a} = (a, a'), \mathbf{b} = (b, b'), \dots$

### 4.3 Multilayer DU2 circuits

In this section, we present multilayer constructions that give rise to DU2 circuits with different entanglement velocities $v_E \neq 1/2$ and discuss their ergodicity properties. The basic construction idea is to take the tensor product constructions of DU2 circuits [35] and couple the tensor factors by using multilayer biunitary connections. For concreteness, we focus on gates constructed from multilayer complex Hadamard matrices.

First, we couple a DU gate (dimension $q_1$) to a DU2 gate (dimension $q_2$) obtained from a kagome lattice. The total gate then inherits the DU2 property of the DU2 factor, as DU2 is a weaker condition on the gate than DU. We show explicitly that this gate satisfies the DU2 condition in Appendix A. When arranged in a brickwork pattern, this gate gives rise to the circuit depicted in Fig. 2. This construction returns a gate with an entanglement velocity $v_E > 1/2$ as

$$U = \qquad \qquad \qquad \text{(95)}$$

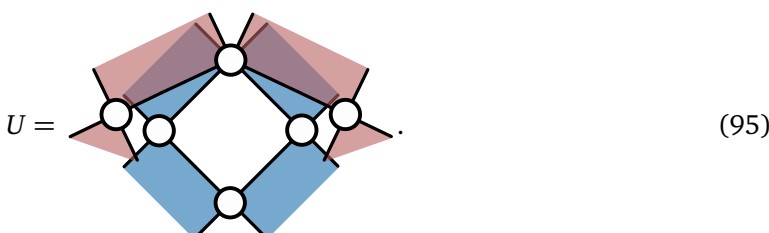

In matrix elements, this construction reads

$$U_{\mathbf{ab,cd}} = H_{a'a,b'b} K_{b'd'} K_{d'c'} K_{c'a'} \tilde{K}_{bd} \tilde{K}_{ac}/(q_2 q_1) \,, \tag{96}$$

where $H$ is taken to be constructed by coupling a $q_1$-dimensional and a $q_2$-dimensional CHM, and $K$ ($\tilde{K}$) denote general CHMs with dimension $q_1$ ($q_2$).

In this example, only the top CHM couples the layer through the HS construction, while the other two multilayer CHM are taken to be tensor products. However, this is not essential to the entanglement dynamics.

The entanglement velocity can be read off from the Schmidt rank. The rank is $\mathcal{R} = q_1^2 q_2$, where the DU layer contributes $q_1^2$, while the DU2 layer contributes $q_2$. As a result

$$v_E = \frac{\log\left(q_1^2 q_2\right)}{\log\left(q_1^2 q_2^2\right)} \,. \tag{97}$$

The value of the entanglement velocity can also be motivated using the minimal cut picture [71] by noting that the amount of bonds cut differs between the layers, as it depends on the circuit geometry.

Another way to generate a multilayer DU2 gate is to couple a DU gate to a product of one site gates

$$U = \qquad \qquad \qquad \text{(98)}$$

In matrix elements, this construction reads

$$U_{\mathbf{ab,cd}} = H_{a'a,c'c} H_{b'b,d'd} K_{ab} K_{cd}/(q_1 q_2) \,, \tag{99}$$

where $H$ now couples a $q_2$-dimensional one-site gate to a $q_1$-dimensional CHM.

The entanglement velocity of these gates is given by

$$v_E = \frac{\log\left(q_1^2\right)}{\log\left(q_1^2 q_2^2\right)}. \tag{100}$$

Finally, we couple a kagome lattice DU2 gate to a product of one site gates

$$U = \qquad \tag{101}$$

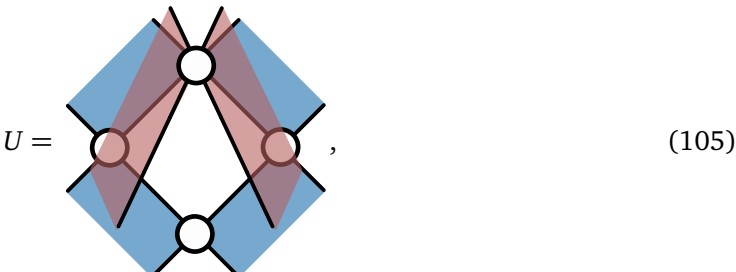

such that $U$ is expressed in terms of $q_1$-dimensional CHMs coupled to $q_2$-dimensional one-site gates as

$$U_{\mathbf{ab,cd}} = K_{ac} H_{b'b,d'd} H_{a'a,c'c}/(q_1 q_2), \tag{102}$$

yielding an entanglement velocity

$$v_E = \frac{\log\left(q_1\right)}{\log\left(q_1^2 q_2^2\right)} < 1/2. \tag{103}$$

## 4.4 Multilayer DU3 circuits

In this section, we extend the multilayer constructions to obtain DU3 gates. There are two known classes of gates that satisfy the DU3 condition and can be obtained from biunitary connections: diagonal gates and "sheared dual-unitary" gates [35,37]. The former are strongly non-ergodic, while the latter only satisfy one of the two DU3 conditions, corresponding to one of the two light-cone directions in spacetime. Here, we couple diagonal gates to DU or DU2 gates using the multilayer approach. This yields more general instances of DU3 dynamics that nevertheless possess a notion of solvability stronger than pure DU3 circuits. We show that such circuits remain partially non-ergodic.

Diagonal gates that cannot be represented in product form are the simplest examples of gates satisfying the DU3 condition. A simple way to ensure that the gate is not a tensor product is to choose the diagonal entries from a complex Hadamard matrix,

$$D_{ab,cd} = \delta_{ac}\delta_{bd}H_{ab}. \tag{104}$$

We generate non-trivial DU3 gates by coupling such a diagonal gate to a gate belonging to a lower level of the hierarchy, for example to a DU gate

$$U = \qquad , \tag{105}$$

resulting in

$$U_{\mathbf{ab,cd}} = H_{a'a,b'b}\delta_{ac}\delta_{bd}K_{b'd'}K_{d'c'}K_{c'a'}/q_1, \tag{106}$$

or to a DU2 gate

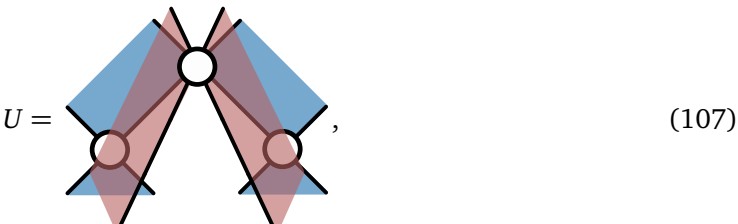

$$U = \qquad\qquad\qquad , \tag{107}$$

for which

$$U_{\mathbf{ab,cd}} = H_{a'a,b'b}\delta_{ac}\delta_{bd}K_{a'c'}K_{b'd'}/q_1 \,. \tag{108}$$

In both instances, the complex Hadamard matrix on top is generated from the HS construction.

Gates generated from the above constructions are not fully ergodic. To see this, consider an initial product state in the computational basis. This state also factorizes between the layers. The state in the top layer is now left invariant (up to an overall phase) by the action of the diagonal gate

$$U(|ij\rangle \otimes |kl\rangle) = (U^{kl}|ij\rangle) \otimes |kl\rangle \,. \tag{109}$$

The product state in the top layer acts as a control for the DU (DU2) circuit in the bottom layer. Thus, such initial states never thermalize in the top layer and never get entangled between the layers. This observation enables us to exactly characterize the dynamics in a large class of initial states: those that can be expressed as a product of a solvable state in the bottom layer and a computational basis state in the top layer. As the top layer only acts as a control and does not contribute to the entanglement growth, all the entanglement is produced in the bottom layer, yielding an entanglement velocity of

$$v_E = \frac{\log\left(q_1^2\right)}{\log\left(q_1^2 q_2^2\right)}, \quad \text{DU} \otimes \text{Diag} \,, \tag{110a}$$

$$v_E = \frac{\log(q_1)}{\log\left(q_1^2 q_2^2\right)}, \quad \text{DU2} \otimes \text{Diag} \,. \tag{110b}$$

# 5 Conclusion and outlook

In this work, we investigated multi-unitary circuit dynamics for which the dynamics of correlations and entanglement remains exactly solvable. Through the use of biunitary connections, we presented a purely graphical construction that unifies constructions of hierarchical dual-unitary (DU2) and triunitary models by arranging biunitary connections on the kagome lattice. Using biunitarity, it can be directly shown that correlation functions of these models are only nonvanishing on three isolated rays in spacetime satisfying $x = 0$ and $x = \pm t$. Furthermore, following a quantum quench from initial solvable states, these models thermalize exactly after a finite number of time steps. By identifying different unit cells for the kagome lattice, this class of circuits can be viewed both through the lens of triunitarity or DU2. We outlined a general framework for the construction of circuits with DU2- and triunitary-type dynamics out of various biunitary connections. These constructions return both known and novel families of DU2 and triunitary models, and naturally lead to new notions of multi-unitary such as triunitarity round-a-face.

Entanglement growth in DU2 circuits, as quantified through the entanglement velocity, is known to be quantized. We here additionally presented a scheme for constructing such circuits with all possible known entanglement velocities. These constructions generalize previous

tensor product constructions through the use of multilayer biunitary connections. The resulting circuits break the full hexagonal symmetry of previous constructions, thus enabling the appearance of an entanglement velocity $v_E \neq 1/2$.

Our results suggest a close connection between geometric structures in spacetime and the solvability of quantum circuits defined on them. We have shown that for a variety of constructions, DU2 and triunitary solvability can be traced back to biunitarity, but with the biunitary connections defined on the kagome lattice and on certain extensions. It remains an open problem to find more geometries leading to solvable dynamics of a potentially more general type. Is it possible to find a unifying principle behind such "geometric" solvability and can it always be expressed in terms of local conditions on few gates? It would also be interesting to explore random or quasiperiodic geometries (c.f. Ref. [72] for dual-unitary circuits defined on the intersections of randomly placed lines) as well as higher spacetime dimensions.

Furthermore, random matrix-like spectral correlations have been demonstrated analytically only in the lowest level of the hierarchy, dual-unitary circuits. It would be interesting to study signatures of random-matrix-like behavior, such as spectral form factors, quantum state designs, or inverse participation ratios, in more complex solvable models.

## Acknowledgments

We acknowledge useful discussions with Chuan Liu, Jiangtian Yao, and Xie-Hang Yu.

## A   Generalized dual-unitarity of multilayer constructions

In this section, we show that the family of multilayer gates given by Eq. (95) satisfies the DU2 condition. The proof carries over analogously to the other constructions presented in Sec. 4.3 and Sec. 4.4 (regarding the DU3 condition).

The left hand side of the DU2 condition is graphically expressed as

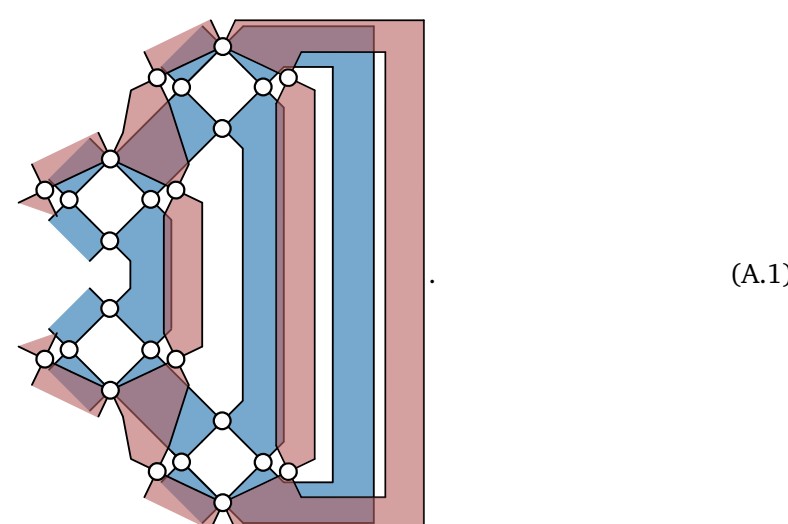

$$\tag{A.1}$$

In the first step, we use the biunitarity properties of the single-layer CHM to reduce the diagram to

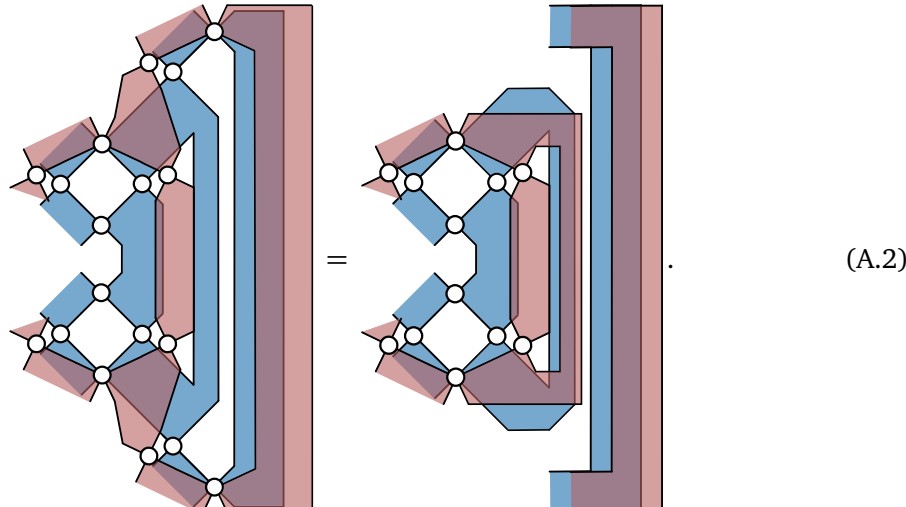

$$\hspace{10cm} (A.2)$$

In the final equality we have applied horizontal unitarity to the multilayer CHM connecting both layers. This final expression yields the identity in both layers separately, and the DU2 condition follows.

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
