# Peer review of "Geometric constructions of generalized dual-unitary circuits from biunitarity"

_SciPost Physics, doi:SciPost Phys. 18, 182 (2025)_

## Round 1 · Referee Report · Anonymous (Referee 1) · 2025-1-9

Strengths

1-Novel Results on Dual Unitary circuits, 2-clear and pedagogical presentation of the results

Report

The paper with title "Geometric constructions of generalized dual-unitary circuits from biunitarity" presents a framework where with the help of biunitarity and shaded calculus, generalise dual unitarity to different geometric constructions.
In section 2, the authors offer a brief introduction to notions like dual-unitary and triunitary gates, their hierarchical extensions (DU2, DU3), biunitarity and shaded calculus .
In section 3, they present how the biunitary connections and shaded calculus can be used to lead to new dual unitary and triunitary geometries, with exact results on the entanglement velocities of the circuits.
In section 4, they use multilayer biunitary connections to construct generalisations of DU2 and DU3 and they propose a mehtod of achieving different entanglement velocities , from the specifics of the construction of the gates.

In general, I believe that this work offers, novel results on finding solvable many-body quantum systems and a framework that can open new pathways of research to not just the Kagome lattice, but to other geometries as well. The manuscript is pedagogically structured, with brief introductions to notions that the reader needs, thus making the presentation of the results clear and accessible to a wider range of researchers. Based on the above points, this manuscript satisfies the acceptance criteria of the journal and I only have to recommend a few minor changes (mentioned in the list below).

Requested changes

1- In p.2 : "Dual-unitary circuits were first..." -->Dual unitarity circuits are also examples of non chaotic models.

2- In p.2: " ...certain aspects of their dynamics pathological"--> Can you be more explicit about which aspects are pathological?

3- In p.4: "Multiple works extended the notion..."--> I would suggest to cite some relevant literature.

4- In p.21 --> I would suggest cleared indication of the time and space axis and of the bipartion $A,\bar{A}$

Some general comments
5- Since the Schmidt Rank is mentioned multiple times, I would suggest a more explicit explanation of it.

Recommendation

Ask for minor revision

---

## Round 1 · Referee Report · Anonymous (Referee 2) · 2025-1-30

Strengths

- The manuscript provides a systematic organization of various generalizations of dual-unitary circuits within a unified mathematical framework.

- The use of the shaded calculus extends tensor network representations, allowing for a compact encoding of unitarity constraints.

- The demonstration that different unitary circuit constructions (triunitary models and hierarchical DU2/DU3 structures) naturally emerge from the same underlying framework is a key result.

- The paper introduces explicit methods for constructing generalized gates with tunable (though quantized) entanglement velocity, broadening the scope of exactly solvable models.

- The study is timely and relevant, offering an important organizational perspective on recent developments in exactly solvable quantum circuit models.

Weaknesses

- The discussion of unitarity conditions using the shaded calculus is concise but may be difficult for readers unfamiliar with this formalism.

- Section 3.4 is technical and lacks an introductory explanation outlining the motivation behind classifying different constructions.

- The multilayer constructions in Section 4 remain conceptual, with limited explicit examples illustrating their organization into many-body quantum circuits.

- Some graphical relations (Eqs. 13-17) are presented without explicit algebraic equations, which could make it harder for readers to follow the formalism in practice.

Report

The paper is well-written and provides a strong conceptual framework for unifying various generalizations of dual-unitary circuits. The use of the shaded calculus is particularly interesting, as it extends the familiar tensor network representations to a more flexible structure that incorporates geometric regions corresponding to tensor contractions. The authors successfully use this formalism to encode different unitarity constraints compactly and then apply it to construct structured many-body circuits on the Kagome lattice. The demonstration that different unitary circuit constructions (triunitary models and hierarchical DU2/DU3 structures) naturally arise from the same underlying framework is a key result of the paper.

However, despite the elegance of the formalism, some aspects of the exposition could be improved to enhance clarity, particularly for readers unfamiliar with the shaded calculus. In particular:

The discussion of horizontal and vertical unitarity conditions in terms of shaded regions (Eqs. 11 and 12) is compact and efficient but would benefit from explicit examples where known special cases are derived by selecting specific colorations of the diagram. Providing explicit algebraic equations corresponding to graphical relations (especially in Eqs. 13-17) would greatly help readers unfamiliar with the notation. In particular, it was not clear to me at first the need for colouring regions (instead of just the tensor) with different colours and what happens when the same region is split into two different colours.

Section 3.4, which surveys various explicit constructions of dual-unitary gates and their extensions, is highly technical and somewhat difficult to follow. While the motivation for classifying these different constructions is clear, a more structured introduction to this section would be beneficial. Specifically, it would help to outline why different approaches (dual-unitary gates, complex Hadamard matrices, etc.) are useful and what properties they introduce to the circuit models.

Section 4, discussing multilayer constructions, extends the biunitary connection formalism to non-planar diagrams. While this is a natural and intriguing extension, the discussion remains largely conceptual. Given the novelty of this approach, it would be useful to provide at least one concrete example illustrating how these structures can be explicitly organized into many-body quantum circuits. At present, the discussion is predominantly textual, making it difficult for readers to intuitively grasp the resulting circuit behavior.

Overall, the paper presents a compelling and well-motivated study that successfully unifies different generalizations of dual-unitary circuits within a common mathematical framework. With the requested clarifications, it will be an even stronger contribution to the field.

Requested changes

- Expand the discussion of unitarity conditions in terms of the shaded calculus by providing explicit examples of how known cases emerge from specific choices of colored regions. Converting the graphical relations in Eqs. 13-17 into explicit algebraic equations would be particularly helpful, also to clarify the meaning of joining regions of different colours.

- Improve the readability of Section 3.4 by including a brief introductory paragraph outlining the motivation for the different constructions and their physical significance.

- In Section 4, provide at least one explicit example of a multilayer construction showing how these structures organize into many-body quantum circuits. This would make the discussion more concrete and accessible.

Recommendation

Ask for minor revision

---

## Round 2 · Referee Report · Anonymous (Referee 1) · 2025-3-19

Report

The authors made the appropriate changes and I believe that the manuscript can be published as it is. I would like to thank the authors for their work and changes.

Recommendation

Publish (meets expectations and criteria for this Journal)

---

## Round 2 · Referee Report · Anonymous (Referee 2) · 2025-5-9

Report

The new version of the manuscript has been improved in clarity in the points I listed in my first report. I am happy to authorize publication of the article in its current form.

Recommendation

Publish (meets expectations and criteria for this Journal)

---

## Round 2 · Author Response

Dear Editor,

please find attached our resubmission of "Geometric constructions of generalized dual-unitary circuits from biunitarity" by Michael Rampp, Suhail Rather, and Pieter Claeys.
We are grateful to the referees for their constructive and helpful suggestions. We have incorporated these suggestions into the revised manuscript.

Regards,
Michael Rampp (on behalf of all authors)

---

## Round 2 · List of Changes

• changed wording of introduction and added examples of pathological behavior of dual-unitary circuits
  • Sec. 2.2: added references to literature
  • Sec. 2.2: added an expanded introduction to the Schmidt rank
  • Sec. 2.3: added explicit discussion of vertical and horizontal unitarity of several biunitary connections with graphical and algebraic equations, including explanation of the necessity of shaded regions
  • Sec. 3.4: added a motivating paragraph contextualizing the various constructions
  • Sec. 3.4.5: included graphical indication of bipartition in Eq. (75)
  • Sec. 4: improved motivation of tensor product constructions
  • Sec. 4.1: added explicit examples of multilayer complex Hadamard matrices that can be used to construct multilayer circuits
  • Sec. 4.2: added Fig. 2 depicting the circuit diagram of a multilayer circuit
  • Sec. 4.2 and 4.3: added algebraic expressions for multilayer gates

---

## Editorial Decision

published